# The Role of Plasmodesmata-Associated Receptor in Plant Development and Environmental Response

**DOI:** 10.3390/plants9020216

**Published:** 2020-02-07

**Authors:** Minh Huy Vu, Arya Bagus Boedi Iswanto, Jinsu Lee, Jae-Yean Kim

**Affiliations:** 1Division of Applied Life Science (BK21 Plus Program), Plant Molecular Biology and Biotechnology Research Center, Gyeongsang National University, Jinju 660-701, Korea; vuhuyminh@gnu.ac.kr (M.H.V.); leejinsu@gnu.ac.kr (J.L.); 2Division of Life Science, Gyeongsang National University, 501 Jinju-daero, Jinju 52828, Korea

**Keywords:** plasmodesmata, receptor-like protein, receptor-like kinase, environmental stresses, plant development

## Abstract

Over the last decade, plasmodesmata (PD) symplasmic nano-channels were reported to be involved in various cell biology activities to prop up within plant growth and development as well as environmental stresses. Indeed, this is highly influenced by their native structure, which is lined with the plasma membrane (PM), conferring a suitable biological landscape for numerous plant receptors that correspond to signaling pathways. However, there are more than six hundred members of *Arabidopsis thaliana* membrane-localized receptors and over one thousand receptors in rice have been identified, many of which are likely to respond to the external stimuli. This review focuses on the class of plasmodesmal-receptor like proteins (PD-RLPs)/plasmodesmal-receptor-like kinases (PD-RLKs) found in *planta*. We summarize and discuss the current knowledge regarding RLPs/RLKs that reside at PD–PM channels in response to plant growth, development, and stress adaptation.

## 1. Plasmodesmata-RLPs/RLKs

Plant cells have remarkably evolved symplasmic channel structures, Plasmodesmata (PD), to create a bridge for cell-to-cell transport of essential molecules such as water, ions, nutrients, phytohormones, and macromolecules including RNAs and proteins [1,2,3,4,5,6,7,8,9]. In general, PD structures are composed of three major components: a plasma membrane (PM), a cytoplasmic sleeve (CS) and an endoplasmic reticulum (ER) called desmotubule (D) [10,11]. By controlling intercellular trafficking of numerous essential factors, PD are indeed particularly linked to plant growth and development as well as playing a critical role in the response to abiotic and biotic stresses [12,13,14,15,16,17,18,19,20,21,22,23,24,25,26]. The existence of PM and PM-lipid raft at the PD channels [27,28] provides a suitable platform for cell surface receptors to be localized and/or relocalized at PD in response to either developmental or environmental-related stimuli [23,29,30,31,32]. These cell surface PD receptors include receptor-like proteins (RLPs) and receptor-like kinases (RLKs) that possess distinct extracellular domains (such as cysteine-rich domains, leucine-rich repeat domains or lysin motifs) to relay intracellular signaling [15,21,22,29,32,33,34,35,36,37,38,39,40,41]. Typical PD-RLPs contain the unique extracellular ligand-binding domain, a single transmembrane domain and a short cytoplasmic tail [21,36,42]. In another case, a PD-RLP uses a glycosylphosphatidylinositol (GPI) anchor to attach the extracellular membrane instead of transmembrane domain [23,43,44]. Meanwhile, PD-RLKs carry out an extracellular domain, a single transmembrane domain, and an intracellular kinase domain [32,43]. In general, extracellular domains of RLKs/RLPs recognize the ligands with high specificity and selectivity; they subsequently modulate the activation of the cytoplasmic kinase domain and the downstream signaling cascades [45]. In *Arabidopsis thaliana*, several membrane-localized receptors have been identified, and some of them localize to PD (Table 1.) Moreover, in other plant species such as *Oryza sativa* and *Populus trichocarpa*, some PD receptors have been reported, but their roles are still elusive [46,47,48]. 

## 2. Abiotic Stress-Involved PD-RLKs

Plants are challenged by many environmental stresses including abiotic and biotic stress. Therefore, plants have advanced sophisticated recognition systems to detect environmental stimuli mediated by cell surface/membrane-localized receptors. Biologically, plant receptors perceive the ligands, subsequently transduce the extracellular signals to the downstream signaling of the receptor complexes through activation of phosphorylation events [35,45,49]. These phosphorylation occasions are the key signaling modules for regulating diverse cellular and physiological responses to establish the proper plant growth, development, and defense responses against various environmental conditions [50,51]. Abiotic stress is defined as a negative effect caused by non-living factors which are often encountered by plants such as extreme levels of light, radiation (UV–B and UV–A), low (cold/chilling/freezing) or high temperature (heat), flooding, submergence, drought, chemical factors (aluminum, arsenate, cadmium, and pH), excessive salt in the soil, deficient or excessive macro/micronutrients, gaseous pollutants (ozone, sulfur dioxide, etc.), and other abiotic factors [52,53]. Moreover, drought and salinity are prominent abiotic stressors with a serious and detrimental impact on plant development as well as agricultural yield productivity [54,55,56,57]. These two abiotic stressors have been fundamentally linked to plant hormonal pathways, which is abscisic acid (ABA), a plant phytohormone designated as a key regulator in the activation of osmotic stress-responsive genes upon drought and salinity conditions [58,59,60]. Additionally, there are several leucine-rich repeat receptor like-kinases (LRR-RLKs) that have been proven to be involved in the response to drought- and salinity-activated ABA signaling pathways, however, most of them are localized at the PM compartment [61,62,63,64,65,66,67,68]. A recent report showed that two LRR-RLKs of *Arabidopsis thaliana*, QSK1 (Qian Shou kinase) and IMK2 (inflorescence meristem kinase 2) localized in the PM upon the normal condition, but this PM-located QSK1/IMK2 is phosphorylated and subsequently relocalized at PD–PM channels in response to salt and mannitol treatments [32]. QSK1 plays a key role in lateral root (LR) formation by regulating callose deposition upon mannitol treatment [32], but the biological function of IMK2 remains to be identified. In addition to salinity and mannitol conditions, a cysteine-rich receptor-like kinase 2 (CRK2) mainly localizes to the PM under standard growth conditions, but in the presence of excess salt and mannitol conditions, this protein is accumulated at PD–PM and required for salt-induced callose deposition (Table 1.) [69]. The formation of callose at PD is induced by environmental stimuli, and the emergence of these PD-RLKs in response to abiotic factors provides a key attention to uncovering the biological mechanisms that detail the unanswered questions which need future research to be answered.

## 3. Biotic Stress-Involved PD-RLPs/RLKs

Biotic stress in plants is defined as a negative impact of living organisms (including pathogens), specifically viruses, bacteria, fungi, nematodes, or insects. Plants also have various chemical and physical defense layers to protect themselves from pathogens. In terms of callose related to physical defense, the most influential physical resistance, Powdery mildew resistant4 (PMR4) or Glucan synthase-like 5 (GSL5) are the rapid response of callose deposition on plasmodesmata to powdery mildew in the papillae formation [85]. On the other hand, to perceive the pathogens and herbivores, plant immunity relies on innate immune receptors expressed in each cell, which recognize invasion signals to mount pattern-triggered immunity (PTI) or effector-triggered immunity (ETI) (The plant immune system). PTI is the first active defense layer of the plant immune system and can be considered as the basal resistance of interaction between plants and microbes via the recognition of conserved pathogen- or microbe-associated molecular patterns (PAMPs/MAMPs) by plant pathogen- or pattern-recognition receptors (PRRs). Recently, PD-callose homeostasis regulation has been reported to be a non-cell-autonomous process regulated by pathogen perception defense or an immune response activated by PAMPs [23,30,35]. 

In the case of plant fungal-triggered PTI response, it has been reported that PAMPs chitin could trigger a reduction in the PD flux or PD permeability. In chitin response, the receptor-like protein LYSIN MOTIF DOMAIN-CONTAINING GLYCOSYLPHOSPHATIDYLINOSITOL-ANCHORED PROTEIN (LYM2) is employed to increase callose deposition upon *Botrytis cinerea* infection [23]. LYM2 locates to PM and PD but the mechanism of LYM2-mediated plasmodesmal closure remains unknown. Recently, Faulkner’s group provided the evidence that LYK4 and LYK5 (LysM-CONTAINING RECEPTOR-LIKE KINASE4 and 5, respectively) were also involved in response to chitin-triggered plasmodesmal closure, raising the question of how LYM2, LYK4, and LYK5 integrate to regulate PD permeability in response to chitin [43]. However, based on the subcellular localization study, LYK5 and LYK4 are mainly localized to PM at the steady condition and only LYK4 is strongly accumulated at PD–PM in the presence of chitin. Although LYK5 is not located at PD–PM in the absence or presence of chitin, this protein still has a function in the chitin-triggered plasmodesmata closure by regulating the LYK4 function. Phosphorylation of LYK4 is necessary for PD–PM-relocalized LYK4 upon chitin-triggered plasmodesmata closure. The modification of LYK4 by LYK5 does not occur in the PD and it most likely happens in the PM. Faulkner’s group mentioned that the mechanism of RLKs response to chitin required the complex formation of family proteins. It also suggested that the reactive oxygen species (ROS) burst together with calcium wave downstream of the LYM2-mediated chitin, signaling that pathway plays the key role in callose accumulation for plant innate immune systems. Subsequently, the PD–PM relocation event of cell surface RLKs takes place presumably in order to perceive the ligand from the pathogens and then regulate callose deposition at PD. Moreover, LYM2 is required to induce callose accumulation in response to chitin but not to flg22 response [23]. Nonetheless, flg22 peptide-triggering callose deposition has been well studied [86,87,88]. 

ROS perception or signaling transmission is recognized by the cysteine-rich receptor-like kinases (CRKs) [44]. CRKs are one of the largest groups of RLKs in Arabidopsis with 44 members [89]. CRK2 is identified as a receptor that contains duplicated domains of the unknown function 26 (DUF26) structure C-X8-C-X2-C. Moreover, it has been reported that CRK2 also relocalized from PM to PD under salt stress [69]. In a *crk2* mutant plant, the callose level is attenuated compared with a wild type plant in response to an excessive salt condition. In terms of biotic stress, *crk2* a mutant plant is susceptible to *Pseudomonas syringae pv.* Tomato DC3000 is an avirulent bacterial pathogen, indicating that CRK2 is involved in the PTI response [75]. Even though Ca^2+^ cytosolic signaling is reduced in the absence of *CRK2*, flg22-dependent MAPK activation is rapidly increased. This is similar to CML41, in which *CRK2* acts independently of the ROS generation to regulate the accumulation of callose in response to flg22. In the downstream pathway, flg22-induced callose is linked to GSL5 function [74] and CALMODULIN-LIKE PROTEIN 41 (CML41) [75]. CML41 functions specifically in response to bacterial flg22, but not fungal chitin. However, the downstream signaling pathways connecting the CML41 protein and the regulation of callose turnover have not been elucidated. It has been reported that CRK2 facilitates MAPK activation and negatively regulates callose deposition through GSLs after MAMP recognition. 

On the other hand, the PD LOCATED PROTEIN (PDLP) family consists of eight receptor-like proteins that contain a cytoplasmic domain, a single transmembrane domain and two extracellular DUF26 (specific targeting of a plasmodesmal protein affecting cell-to-cell communication). PDLP1 and PDLP5 (originally named HOPW1-1-INDUCED GENE1 (HWI1)) are functionally designated to biotic stress such as fungal, virus, and bacterial pathogens. It has been reported that PDLP5 interacts with GSL8 but the mechanism and function remain unknown [90]. At any rate, PDLP5 may induce callose deposition through physical interaction with GSL8 to form the PDLP-GSLs protein complex.

PDLP5 confers a resistance phenotype upon *P. syringae* infections through interacting with a mechanism for salicylic acid (SA)-induced plant immunity. [91]. Consistent with the function of PDLP5 in this response, the authors demonstrated that PDLP5 is induced by a *P. syringae pv maculicola* (*Pma*) infection and plays a critical role in regulating PD continuity [21]. Furthermore, the activation of SA-mediated PD closure requires the action of PDLP5 during bacterial infection [14]. In addition, GSL4 also works together with PDLP5 to maintain basal callose levels at PD but does not require PDLP5 for ROS-dependent plasmodesmal regulation. It can be assumed that the CRKs or other PD-RLKs/RLPs may interact directly or indirectly with GSL4 to regulate ROS-dependent plasmodesmal regulation [92].

It has been postulated that *PDLP5* enhances plant tolerance against the fungal-wilt pathogen *Verticillium dahlia* and bacterial *P. syringae pv* tomato DC3000 (*Pst* DC3000) in the absence of sphingolipid long-chain base Δ8 desaturase (SLD) 1 and 2, whereas *sld1 sld2 pdpl5* triple mutant enhances the plant susceptibility [77]. In the *sld1 sld2* double mutant, t18:0-based sphingolipids are elevated and a PDLP5 expression is induced in the leaf epidermal cells. It has been reported that the accumulation of PDLP5 in *sld1 sld2* double mutant is particularly caused by the specific interaction of phytosphinganine t18:0 with a sphingolipid binding motif at the C-terminus domain of PDLP5. In plants, free d18:0 acts as a second messenger-triggered programmed cell death (PCD) dependent on cytosolic calcium [93]. Therefore, t18:0 might also act as a signaling molecule to elevate PDLP5 expression. Furthermore, the latest discovery demonstrated that the bacteria effector of *Pst* DC3000, HopO1-1, physically interacts with PDLP5 and PDLP7 in arabidopsis. Together, double mutants of these genes showed similar susceptibility to bacterial infection, suggesting that PDLP5 and PDLP7 are required for pathogen immunity. Finally, it is interesting to speculate about PD callose regulation in biotic stimuli, in which PDLP members and the other sphingolipid compositions or lipid raft components are associated to maintain plant growth and development upon biotic stress [27,28]. 

PDLP5 plays a key role in the systemic acquired response (SAR), where PDLP5 interacts with PDLP1, then recruits the AZA1 protein to form a protein [94]. This protein complex regulates the SAR pathway via glycerol-3-phosphate (G3P) and azelaic acid (AzA) [15]. PDLP1 and PDLP5 are essential for SAR as well as for the stabilization of the lipid transfer-like protein AZI1, a key SAR molecule. The loss-of-function of either *PDLP1* or *PDLP5* induces the chloroplastic relocalization of AZI1, a similar pattern to pathogen infection by Pst DC3000. Moreover, it has been reported that PDLP1 is not essential for the basal plasmodesmal permeability even when located at the PD [42]. However, upon the downy mildew pathogen *Hpa* infection, PDLP1 rapidly interacts with SNARE VAMP721 (vesicle-associated membrane protein) to elevate callose accumulation [24]. Additionally, the *pdlp1,2,3* triple mutant is susceptible to *Hpa* infection by reducing callose deposition around the haustoria and host membrane, suggesting that PDLPs are involved in the basal immunity-mediated callose accumulation. In cotton species, PDLP1 and PDLP6 have been proposed to regulate callose accumulation through the SA-dependent transcriptional pathway in response to *Verticillium dahlia* [79]. 

To attack a *host plant*, viruses favorably target PD channels to spread out the viral genomes by modulating the size exclusion limit (SEL). It has been remarkably postulated that the movement proteins (MPs) are encoded by the tobacco mosaic virus (TMV) and the fungal pathogen *Fusarium oxysporum* modify PD SEL [95,96]. Through the open PD, MP-RNA genome or effectors such as Avr2 move from infected cells to the adjacent cells. Furthermore, a cell surface PD receptor-like kinase, BARELY ANY MERISTEM 1 (BAM1) acts in the cell-to-cell movement of RNAi via PD channels through physical interaction with a C4 protein from the tomato yellow leaf curl virus (TYLCV). However, the *bam1* single mutant does not interfere with the intercellular spread of RNAi, only the *bam1 bam2* double mutant exhibits cell-to-cell RNAi movement suppression, indicating BAM1 and BAM2 play a redundant function in this mechanism. A recent study on the BAM1 and BAM2 revealed that these two proteins are required for cell-to-cell movement of miR165/6 to regulate xylem patterning in the Arabidopsis root [38,71]. 

## 4. PD-RLKs Govern Plant Growth and Development

Growth is considered one of a living being’s most basic and recognizable characteristics. Growth can be described as a permanent irreversible increase in the size of an individual cell, tissue, organ or organism. Growth is typically followed by metabolic (both anabolic and catabolic) processes. Plant growth is remarkable because some plants can grow unlimitedly during their lives. This ability of plants is due to the presence of meristems in their bodies at certain places. Plant development can be defined as a cycle of processes from the beginning of the plant component to its death (germination of the seed to senescence). Plant development encompasses both growth and differentiation with quantitative and qualitative changes [97]. In addition, plants use a range of PD–PM receptors to sense endogenous and exogenous signals for plant growth stimulation and development. These PD–PM receptors include the leucine-rich repeat (LRR) receptor kinase CLAVATA1 (CLV1) and the non-LRR receptor kinase ARABIDOPSIS CRINKLY4 (ACR4), which are required for shoot and root stemness maintenance in arabidopsis, respectively [40,70,72,73,74,98,99,100]. Moreover, tissue morphogenesis is another key factor in the biological process during plant growth and development, which usually involves an alteration in cell number, size, shape, and position. These alterations are particularly achieved through several cellular mechanisms such as cell proliferation, cell elongation, and cell-to-cell communication [101,102]. In particular, in PD, the gates of cell-to-cell communication are occupied by the atypical leucine-rich repeat receptor-like kinase (LRR-RLK) STRUBBELIG (SUB), which plays a pivotal role during tissue morphogenesis in Arabidopsis. It has been reported that SUB localizes to PD–PM and plants lacking *SUB* activity show severe defects at plant growth and developmental stages such as floral patterning, stemness maintenance, plant height, and root hair formation [29,39,80,81,103,104,105]. To encourage the optimal growth and developmental processes, plants often cooperate with other living organisms, such as microorganisms (archaea, protists, bacteria, and fungi). These mutually beneficial interactions between two living organisms are often called symbiosis, which involve multidirectional changes in the genome, metabolism, and signaling network. However, plant-microbe interactions can be either beneficial or harmful to one another [106]. The most common study in the beneficial plant-microbe interaction comes from leguminous plants and one of the Rhizobia species. This interaction results in the formation of a root nodule, wherein rhizobia reside and actively fix nitrogen that is used directly by the host plant. Furthermore, to maintain the symbiotic balance between the host plants and rhizobia, negative feedback systems known as autoregulation of nodulation (AON) have evolved in plants. AON inhibits the number of root nodules through short- and long-distance signaling via shoot–root communication and is particularly mediated by an LRR-RLK SUNN (SUPER NUMERARY NODULES) localized to PD–PM in the *Medicago truncatula* (*Mt*) plant [33,34,41,82,83,84,107,108].

## 5. Conclusions

In summary, several PD-RLKs/RLPs have been characterized in plants, mostly in *A. thaliana* (Table 1), but questions remain about their functions in PD regulation. In response to plant development along with environmental stresses, PD-RLKs/RLPs rapidly relocalize from the PM to PD–PM apertures and subsequently stimulate the callose accumulation. Which-type ligands (for example, ROS-like chemicals, chitin-like oligosaccharides, and flagellin-like peptides) are involved, but how these proteins recognize and sense the ligand and interact with their substrates involved in the downstream signaling pathways remain elusive. Recent proteomics-based approaches such as PD proteomic analysis and proximity-dependent biotin identification (Bio-ID) may provide a platform to identify and characterize the new PD-RLKs/RLPs and PD-interacting proteins. Additionally, molecular cell biology and molecular genetic approaches will be helpful in gaining insights into the functional aspects. These approaches include genetic analyses of PD-related mutants to understand their role in signaling pathways and amino acid substitution or domain swapping analyses in the ectodomain or intracellular domain to know signal perception or transduction. CRISPR/Cas-based genome editing tools will be useful for generating knock-out mutations in *PD-RLKs/RLPs* of which T-DNA tagging lines are not available [109]. Cryo-electron microscopy (cryo-EM) might provide deep insight on PD structure or PD proteome structure. Callose is a key molecular player in PD regulation. Callose or its degradation derivatives can act as a potential ligand to alarm the status of PD opening or closing. Although the existence of a plant β-glucan receptor carrying a dectin domain was proposed by Fesel and Zuccaro [110], it remains to be tested. The ROS cause the PD callose accumulation, but how ROS are sensed during callose homeostasis is not yet known. Cysteine residues might sense ROS, thus it will be interesting to identify ROS-sensing receptors among RLKs carrying cysteine-rich ectodomain in order to uncover the ROS-mediated PD regulation pathway. Overall, these insights could be used to explore the role of plant PD-RLKs/RLPs in PD regulation regarding the plant growth and development as well as environmental stimuli.

## Figures and Tables

**Table 1 plants-09-00216-t001:** Plasmodesmal-receptor like proteins (PD-RLPs) and plasmodesmal-receptor-like kinases (PD-RLKs) involved in plant development and environmental stimuli.

Gene Name	Type	Organism	Gene ID	Proposed Role	References
*ARABIDOPSIS CRINKLY 4* (*ACR4*)	RLK	*Arabidopsis thaliana*	AT3G59420	Growth and Development	[40,70].
*BARELY ANY MERISTEM 1* (*BAM1*)	RLK	*Arabidopsis thaliana*	AT5G65700	Biotic stress	[38,71].
*CLAVATA1* (*CLV1*)	RLK	*Arabidopsis thaliana*	AT1G75820	Growth and Development	[40,70,72,73,74].
*CYS-RICH RECEPTOR-LIKE KINASE2* (*CRK2*)	RLK	*Arabidopsis thaliana*	AT1G70520	Abiotic stress and Biotic stress	[44,69,75].
*INFLORESCENCE MERISTEM RECEPTOR-LIKE KINASE 2* (*IMK2*)	RLK	*Arabidopsis thaliana*	AT3G51740	Abiotic stress	[28].
*LYSIN MOTIF DOMAIN-CONTAINING GLYCOSYLPHOSPHATIDYLINOSITOL-ANCHORED PROTEIN 2* (*LYM2*)	RLP	*Arabidopsis thaliana*	AT2G17120	Biotic stress	[23].
*LYSIN MOTIF-CONTAINING RECEPTOR-LIKE KINASE 4* (*LYK4*)	RLK	*Arabidopsis thaliana*	AT2G23770	Biotic stress	[43].
*PLASMODESMATA-LOCATED PROTEIN 1* (*PDLP1*)	RLP	*Arabidopsis thaliana*	AT5G43980	Biotic stress	[24,76].
*PLASMODESMATA-LOCATED PROTEIN 2* (*PDLP2*)	RLP	*Arabidopsis thaliana*	AT1G04520	Biotic stress	[24,76].
*PLASMODESMATA-LOCATED PROTEIN 3* (*PDLP3*)	RLP	*Arabidopsis thaliana*	AT2G33330	Biotic stress	[24,76].
*PLASMODESMATA-LOCATED PROTEIN 5* (*PDLP5*)	RLP	*Arabidopsis thaliana*	AT1G70690	Biotic stress	[15,21,77,78].
*PLASMODESMATA-LOCATED PROTEIN 6* (*PDLP6*)	RLP	*Gossypium barbadense*	-	Biotic stress	[79].
*PLASMODESMATA-LOCATED PROTEIN 7* (*PDLP7*)	RLP	*Arabidopsis thaliana*	AT5G37660	Biotic stress	[78].
*QIAN SHOU KINASE1* (*QSK1*)	RLK	*Arabidopsis thaliana*	AT3G02880	Abiotic Stress	[32].
*STRUBBELIG* (*SUB*)	RLK	*Arabidopsis thaliana*	AT1G11130	Growth and Development	[29,39,80,81].
*SUPER NUMERARY NODULES* (*SUNN*)	RLK	*Medicago truncatula* Genotype A17	-	Growth and Development	[33,34,41,82,83,84].

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
