# Peer review of "The Role of Plasmodesmata-Associated Receptor in Plant Development and Environmental Response"

_plants, 2020, doi:10.3390/plants9020216_

Round 1
Reviewer 1 Report
Major comments
The manuscript aimed to review the PD-RLP/RK/RLK progress in planta, but only mentioned the findings in Arabidopsis (mainly) and Medicago (1 example). Need to address (at least briefly) the progress in other plants, such as in rice etc. Table 1 needs to be well organized. The targets are not listed according to protein types (ie. PD-RLPs/RKs/RLKs), or the stimuli (abiotic/biotic)/development classes, even not in the order of mutants shown up in the manuscript. Also, instead of starting from mutants, it would be clearer to start from proteins with their information as RLPs, RKs or RLKs etc. Figure 1 is not very informative or self-explained. a) it would be helpful to turn figure legends into the figure. b) it is not clear at all what are the upstream stimuli/ligands or downstream signals in this figure. c) it is also not clear what are the connections with each other, such as with callose.
Minor comments
Need to briefly introduce the similarity/difference among Receptor Like Proteins (RLPs), Receptor Kinases (RKs) and Receptor-Like Kinases (RLKs). Line 22-25, “However, there are more than six hundred … to respond to the external stimuli.” is not proper shown in abstract and is mis-leading the topic in the context. Several transition words are not properly used causes hard understanding or is mis-leading. Line 43, “distinct extracellular domains”, specify what is the distinct domains. Line 111-114, “In the downstream pathway, flg22-induced callose … CML41 protein to regulate callose turnover.” is not relevant to the context here. Better to remove or place after/around “flg22-dependent MAPK activation is rapidly increased” in line 123 which is relevant. Line 103, “This paper”, are you mentioning ref 72? Ref 72 is in bioRxiv and not considered as a publication yet. Probably “the Faulkner’s group [72]” is more accurate. Line 121, “In the crk2 mutant plant, the callose level is… an excessive salt condition.” is better to go after line 119 “PM to PD under salt stress [62].” as both are talking about abiotic stress. Several references are needed: line 76-77, ref for definition of biotic stress; Line 186, Ref for “Plant development … of the seed to senescence”; Line 241, ref for “Cysteine residues were known to sense ROS”.Author Response
I would like to express my sincere appreciation to the reviewer for positive feedback to improve this manuscript.
Please see the attachment.

Reviewer 2 Report
The review from Minh Huy Vu et al integrates current knowledge on the role of receptor proteins located at plasmodesmata in the plant response to biotic and abiotic stresses as well as during plant development. The review is timely with recent publications of high impact in this field that provide significant advances on the regulation of symplasmic communication by cell surface receptors located at plasmodesmata and its impact on plant development and environmental responses.
The article is comprehensive approaching several aspects of the literature with varying levels of depths. I have several comments that can be sorted with a more careful attention to the writing and organization of the manuscript (see below).
I find table 1 useful but the column on characteristic need to be revised to emphasise the evidence that highlight connections of these receptors to plasmodesmata function. A critical oversight is the lack of gene ID for the SUNN protein (which is the only one mentioned outside the Arabidopsis genome). I think table 1 title should read ‘available mutant lines for receptor kinases associated to plasmodesmata and with a described function in plant development and response to environmental stress’.
The lack of mention in the text to RKs identified in other systems (such as rice by the authors) is also intriguing. I also think the authors should read and revise the information provided in a recent paper regarding DUF26 receptors (Mechanistic insights into the evolution of DUF26-containing proteins in land plants. Vaattovaara A, Brandt B, Rajaraman S, Safronov O, Veidenberg A, Luklová M, Kangasjärvi J, Löytynoja A, Hothorn M, Salojärvi J, Wrzaczek M. Commun Biol. 2019 Feb 8;2:56. )
And also describe how the most recent paper on CLE40 signalling and root stemness (Berckmans B, Kirschner G, Gerlitz N, Stadler R, Simon R. Plant Physiol. 2019 Dec 5. pii: pp.00914.2019.) fits with the evidence of a role for PD in plant development.
and expand on the end of section 4 which finish abruptly without really describing the results reported by Crook et al (paper The systemic nodule number regulation kinase SUNN in Medicago truncatula interacts with MtCLV2 and MtCRN.) or its significance for plant development (the title of this section is PD-RKs govern plant growth and development).
Perhaps the author could remove lines 180-186 which do not provide any new information on the topic in question and it is out of place (this should be part of introduction if decided to keep it).
Finally I have major issues with Figure 2, what information this provides that is not already in table 1? How this figure inform on the mechanisms or on the pathway or on the function of these RKs? How the PD in the SAM and RAM differs from the leaves besides having different receptors? Does the figure implies that leaf PDs and RLKs does not contribute to plant growth and development?
The figure legend is massively uninformative and just give a summary of the information described in the text not the information presented in the figure.
Besides these issues, writing should be revised as there are several grammatical mistakes that makes the text difficult to understand. There are also some formatting issues such as ‘Hyaloperonospora arabidopsidis’ appears bigger than the rest of the text in the figure legend and the same for ‘host plant’ in line 168.
Some others (not exhaustive revision) are detailed below:
-line 35 ‘s’ missing in macromolecules
-line 43 should read ‘receptor-like kinases (RLKs) that possess distinct…’
-table 1 title ‘List of mutants in PD-localized RLP,RK,RLK with a role in plant development and environmental responses’ (or see alternative above). Also in the table gene ID for sunn is missing and in the characteristics of pdlp1,2,3 should read ’… sporulation phenotype in pdlp1 was similar to wildtype but was enhanced in pdlp1,2,3, triple mutant. Pdlp1,2,3 suppresses ??? (mention what type of tubule or if this the Hpa) tubule formation.
- the table should be also cited in section 3 and 4 as description in this section also refer to the table.
- Line 76 correct Biotic stress in plants is defined as the negative impact of living organisms (including pathogens).
- line 96 I suggest to change for ‘ raising the question how LYM2, LYK4, and LYK5 integrate to regulate PD permeability.’
-line 105 ROS should be defined first time use
- line 109 Sentence should finish after reference 23 and start again in 110.
-line 113 I suggest: However, the downstream signaling pathways connecting the CML41
protein and the regulation of callose turnover has not been elucidated.
-line 117 I suggest: ‘…receptor that contains duplicated domains of the unknown function 26 (DUF26) structure C-X8-C-X2-C.
-line 129 what the line between brackets refer to, this is not obvious (specific targeting of a plasmodesmal protein affecting cell-to-cell communication).
-line 134 should read PDLP5 confers resistance phenotype upon P. syringae infection through interacting with a mechanism for salicylic acid (SA)-induced plant immunity.
-line 135 what ‘they’ refers to at the end of this line??
-line 145 should read ‘…are elevated and there is induced PDLP5 expression in the leaf epidermal cells.
-line 154 should read ‘PDLP5 plays a key role in the systemic acquired response (SAR), where PDLP5 interacts with PDLP1, then recruits AZA1 protein to form a protein complex.
-line 160 missing when should read ‘…basal plasmodesmal permeability even when located at the PD’
-line 200 ‘in addition to SUB’ should be eliminated from the sentence.
-Line 201-214. This should be a different paragraph and should be carefully rephrased. How this is linked to plant growth and organ development (the topic of this section?). It finishes abruptly, the crook et al paper could be further exploited to integrate information linking PD localization of SUNN and nodulation.
-line 231: should be rephrased as In summary, several PD-RLKs/RKs/RLPs have been characterized in plants mostly in Arabidopsis thaliana (Figure 1) but questions remains on the mechanisms involved on PD regulation.
Author Response
I would like to express my sincere appreciation to the reviewer for positive feedback to improve this manuscript
Please see the attachment.

Reviewer 3 Report
The review by Vu et al entitled “The cell surface plasmodesmata- receptors: current status in the environmental stress and plant development” summarizes the knowledge on the plasmodesmata localized receptor-like proteins (RLPs) and receptor-like kinases (RLKs). The authors present a table that summarizes the described genes from Arabidopsis (and one from Medicago) encoding for receptor-like proteins or receptor-like kinases that localize to the plasmodesmata and play a role in development or environmental stresses. Starting with abiotic stresses, they describe the function of RLPs/RLKs in biotic interactions and finally in development. Indeed, in the last years these cellular structures attract huge interest of the researchers and number of plant receptors have been shown acting in the specific membrane domains associated with the plasmodesmata. These molecular gates govern cell-to cell communication and the RLPs/RLKs are often found to regulate the properties of these channels.
Undoubtedly this is an interesting topic for the review, but current manuscript requires revisions before the acceptance.
Here are my detailed comments.
The title is a little bit too long, the authors could consider to revise it. For example: “The role of plasmodesmata-associated receptors in plant development and environmental responses” The Abstract can be extended and can better represent the focus of the review. To be consistent- line 23-24: the authors write Arabidopsis thaliana, but later- rice (instead of Oriza sativa). The part 1 is too short and lacks the major information-what are the RLPs and RLKs? What is in common and different between two groups? What the structural properties and plant-specific properties of these proteins? What is known about this specific PD-associated localization of these receptors? In the next part the authors discuss the role of plasmodesmata-localized RLKs/RLPs in abiotic stress responses. Indeed, not much has been shown in regard to abiotic stress, but mentioning of the two central papers in two sentences in the end of the second part is too short in my opinion. I would ask authors to revise this part, giving a major attention to PD-RLKs in abiotic stress responses and asking the emerging questions that remain unanswered in this field. The Table 1 also needs corrections. Instead of the first column, I would suggest to present the gene names (all known names), organism, Gene ID, to which category it belongs (development/biotic interactions/abiotic stress) and the mode of action if known and the reference. In the third part- the authors focus on the negative biotic interactions only (but this could be extended to all types of interactions with other organisms). Also here the writing is chaotic, without logical structure. Maybe authors can divide this chapter to several parts, according to the type of the interacting organism (for example virus-related; fungi-related RLPs/RLKs; plant-bacteria interactions etc). The last part devoted to plant development. Also here I find the writing can be improved. Line 200 for example- it is not clear what the authors state.:” In addition to SUB, …. Sub undergoes endocytosis through clathrin-dependent pathways…” what do they mean? Please carefully revise the text, to make sure that the statements are clear to the reader…. And here later the authors suddenly jump to positive biotic interactions. From my point of view- it belongs better to the previous part. And again, just facts about different receptors are demonstrated, it would be helpful to present some conclusions, interpretation, further questions etc. Figure 1 is graphically very nice, but shows for example that BAM1 is playing a role in environmental stresses and not during development (which is not correct). I also do not understand what is the impact of this figure, while the Table 1 already summarizes the same. The authors can revise the Figure 1 with an additional value/information in this figure that is not present in Table1.Thank you
Author Response
I would like to express my sincere appreciation to the reviewer for providing valuable comments to improve this manuscript.
Please see the attachment.

Reviewer 4 Report
Basically, plasmodesmata (PD) are cytoplasmic bridges unique of plant cells. They are spanning through the cell wall and contain a centrally located Endoplasmc Reticulum modified strand that is known as desmotubule, which is surrounded by a cytoplasmic sleeve. These structures provide cytoplasmic and membrane continuity between connected cells. The important role of PD in stress condition (e.g. biotic and abiotic stress) is well recognized because they control the free diffusion of molecules of different size and small particles (e.g. plant viruses). The oriphice of each plasmodesma is surrounded by a collar, made of callose deposits, that act as molecular sphintere. By increasing the callose deposits the plasmodesma size exclusion limit decreases, whereas by reducing these deposits the trasport is facilitated because the size exclusion limit is increased. Another mode to control the PD transport is represented by the actin cytoskeleton which it is likely to act as fine tuning mechanism.
Hence, a number of proteins, either secreted and non-secreted, are localized at the PD and are implied in regulating the PD permeability.
The review article by Vu et al. titled "The cell surface plasmodesmata-receptors: current status in the environmental and plant development" summarizes the current knowledge of the cell-surface receptors that reside at the PD-plasma membrane in response to environmental stress and plant development. On the whole, I appreciated the manuscipt though there are some minor changes that could improve the review.
1) Table 1 can be implmented by adding an additional column summarizing how each sigle protein affects the PD permeability and by narrowing the "References" column by just indicating the reference's number;
2) throughout the whole manuscript the authors describe and discuss the recruitment and the activation of these receptor proteins to the PD-structures. Though our current knowledge about their shutting down is poor, from the manuscript it does not emerge that this is a gap and as a such might represent one of the future challenge. This point can be argued a little bit more.
Author Response

(The authors gave the same response as above.)
